# Application of the Single Source—Detector Separation Algorithm in Wearable Neuroimaging Devices: A Step toward Miniaturized Biosensor for Hypoxia Detection

**DOI:** 10.3390/bioengineering11040385

**Published:** 2024-04-16

**Authors:** Thien Nguyen, Soongho Park, Jinho Park, Asma Sodager, Tony George, Amir Gandjbakhche

**Affiliations:** Eunice Kennedy Shriver National Institute of Child Health and Human Development, National Institutes of Health, 49 Convent Drive, Bethesda, MD 20892-4480, USA; thien.nguyen4@nih.gov (T.N.); soongho.park@nih.gov (S.P.); jinho.park@nih.gov (J.P.); asma.sodager@nih.gov (A.S.); tony.george2@nih.gov (T.G.)

**Keywords:** near-infrared spectroscopy, breath holding, cerebral tissue oxygen saturation, cerebral tissue oxygenation, spatially resolved spectroscopy

## Abstract

Most currently available wearable devices to noninvasively detect hypoxia use the spatially resolved spectroscopy (SRS) method to calculate cerebral tissue oxygen saturation (StO_2_). This study applies the single source—detector separation (SSDS) algorithm to calculate StO_2_. Near-infrared spectroscopy (NIRS) data were collected from 26 healthy adult volunteers during a breath-holding task using a wearable NIRS device, which included two source—detector separations (SDSs). These data were used to derive oxyhemoglobin (HbO) change and StO_2_. In the group analysis, both HbO change and StO_2_ exhibited significant change during a breath-holding task. Specifically, they initially decreased to minimums at around 10 s and then steadily increased to maximums, which were significantly greater than baseline levels, at 25–30 s (*p*-HbO < 0.001 and *p*-StO_2_ < 0.05). However, at an individual level, the SRS method failed to detect changes in cerebral StO_2_ in response to a short breath-holding task. Furthermore, the SSDS algorithm is more robust than the SRS method in quantifying change in cerebral StO_2_ in response to a breath-holding task. In conclusion, these findings have demonstrated the potential use of the SSDS algorithm in developing a miniaturized wearable biosensor to monitor cerebral StO_2_ and detect cerebral hypoxia.

## 1. Introduction

Hypoxia is characterized by decreased oxygen availability in tissue due to low blood supply or low blood oxygen saturation. Hypoxia includes (1) hypoxemic hypoxia, which is caused by hypoventilation originating from factors such as low oxygen tension at high altitudes, foreign body inhalation, impaired respiratory drive, neuromuscular diseases and an increase in partial pressure of gases other than oxygen; (2) circulatory hypoxia from abnormal cardiac blood flow as seen in left—right shunt; (3) anemic hypoxia due to low hemoglobin concentration; and (4) histotoxic hypoxia, which can be due to causes like cyanide poisoning [1]. Hypoxic brain injury can result from interrupted cerebral blood flow, due to cardiac arrest, strangulation, severe anemia, systemic hypotension, systemic hypoxia [2], or traumatic brain injury [3]. Cerebral hypoxia occurs when cerebral oxygen consumption is larger than oxygen supply due to metabolic needs [4]. Furthermore, this imbalance results from the impaired autoregulation of cerebral blood flow [5,6], which is influenced by mean arterial pressure [7] and intracranial pressure [3]. Hypoxic conditions result in a multitude of involuntary systemic responses to ensure vital organ functioning [8]. Increases in ventilation rate, tidal volume, heart rate, blood pressure, and cerebral blood flow all function in attempting to counteract permanent damage due to hypoxia [8].

The current standard method to monitor patients in an intensive care unit for hypoxia is arterial blood gas analysis (ABG), which requires blood samples to be drawn from a radial artery [9]. However, the accuracy of an ABG test is affected by barometric pressures and temperatures [10]. Furthermore, blood exposure to air [11] can occur during pneumatic tube transport [12], resulting in an increase in the arterial partial pressure of oxygen [11]. Hemoglobin count is another method to determine hypoxia. A high hemoglobin value can indicate secondary erythrocytosis [13,14], which has been associated with hypoxic lung disease [15,16]. Another measure is jugular vein oxygen saturation (SjvO2), which increases when the brain oxygen supply exceeds consumption and decreases when consumption exceeds supply. However, this method only provides regional cerebral blood oxygenation [4,17]. Brain Tissue Partial Pressure of Oxygen (PbtO2) involves the insertion of a microcatheter on the cortex to measure oxygenation as a widely used complement to standard intracranial pressure monitoring [4]. However, it possesses disadvantages such as only providing information about regional ischemia and hypoxia, instead of the entire cortex, posing the risk of injury or infection and requiring significant time to obtain temperature balance after placement prior to data collection [4].

A V/Q scan, also known as a ventilation (V) and perfusion (Q) scan, is an invasive, two-part test involving radiolabeled markers used to indicate the presence of hypoxia [18]. A hypoxia determination is made when there is a V/Q mismatch (low V/Q ratio) as the alveolar oxygen level, also represented as PAO2, is decreased, which also decreases the arterial oxygen level [1,19]. A single-photon emission computer tomography (SPECT) V/Q scan is particularly useful if patients cannot tolerate a CT scan; however, the Planar Imaging V/Q scan can result in a less accurate determination of pulmonary perfusion [18]. Another method is CT pulmonary angiography, which is used for the diagnosis of pulmonary embolism by identifying structures such as the lung parenchyma and pleural spaces [20]. Many disadvantages of this technique exist, including the degree of patients’ ability to hold their breath, which can affect the signal-to-noise ratio, as well as IV access and image timing, which can affect the opaqueness of pulmonary arteries and their quality. 

Pulse oximetry is the most common noninvasive method to diagnose patients with hypoxia. However, various factors that cause pulse oximetry readings to become inaccurate include the presence of carboxyhemoglobin and methemoglobin, dark skin pigmentation [21,22,23,24], nail polish, dyes, noise from bright lights, electromagnetic interference, motion, low perfusion, sensor location [25], oximeter brand [21,22,26,27] and less sensitivity to non-pulsatile flow [28], and disease [29,30]. Multiple studies have shown greater discrepancies between blood oxygen saturation measured with a pulse oximeter and ABG, particularly for lower oxygenation levels [21,22,23,24,29,31,32,33]. Furthermore, tissue thickness variation can cause different pathlength ratios, which lead to inaccurate readings [34]. Near-infrared spectroscopy (NIRS) is another technique which may be used to evaluate the risk of cerebral ischemia and hypoxia [35]. There are different types of NIRS techniques including continuous-wave (CW) NIRS, time domain (TD) NIRS, and frequency domain (FD) NIRS (fNIRS), with CW NIRS being the most widely available and affordable [4]. The advantage of NIRS over the other techniques of cerebral monitoring include its noninvasiveness, ability for continuous monitoring, and high temporal resolution. So far, five CW NIRS devices have been approved for patient use by the US Food and Drug Administration (FDA), including CerOx (Ornim, Inc., Dedham, MA, USA), EQUANOX 7600 and 8004CA (Nonin Medical, Inc., Plymouth, MN, USA), FORE-SIGHT (CAS Medical Systems, Branford, CT, USA), and INVOS (Somanetics Corporation, Troy, MI, USA) [36]. These devices use the SRS method, which requires at least two source—detector pairs to calculate StO_2_.

In our previous study, we derived the single source—detector separation (SSDS) algorithm to calculate StO_2_ using information from one source—detector pair [37]. In addition, we have proved its accuracy in StO_2_ measurement using both simulation and in vivo data. Simulation data were generated using the Monte Carlo simulation and SSDS-based calculated StO_2_ was compared to the actual StO_2_. In vivo data were measured from the prefrontal cortex during a hypercapnia task and SSDS-based calculated StO_2_ was compared to StO_2_ calculated using the SRS method. This study investigates cerebral StO_2_ change in response to cerebral hypoxia and compares the sensitivity of the SSDS-based calculated StO_2_ with the SRS-based calculated StO_2_ in detecting cerebral hemodynamic change induced by hypoxia. We hypothesize that the SSDS algorithm does not only require less resources than the SRS method to calculate StO_2_ but it is also more robust than the SRS method in detecting changes in StO_2_ in response to cerebral hypoxia. In this study, a breath-holding task was used to simulate cerebral hypoxia. The aim of the study was to use the SSDS to develop a miniaturized wearable biosensor to monitor cerebral StO_2_, as well as to detect cerebral hypoxia.

## 2. Materials and Methods

### 2.1. Participants

The study protocol was approved by the Institutional Review Board of the *Eunice Kennedy Shriver* National Institute of Child Health and Human Development on 31 August 2021 (NICHD protocol #21CH0028), which can be found at clinicaltrials.gov (NCT05035420) (accessed on 1 April 2024). Participant recruitment and experiments were performed from 26 May 2022 to 28 September 2023. This study recruited healthy adult volunteers through the Office of Patient Recruitment at the National Institutes of Health (NIH). The inclusion criteria included (1) participant signature and date on the informed consent form; (2) participant willingness for study procedure compliance and availability for study duration; (3) male or female ≥ 18 years old; (4) good general health confirmed by medical history; (5) no sign of upper respiratory symptoms; and (6) normal range of body temperature on the day of experiment (afebrile, temperature < 100.4 °F). The exclusion criteria for participants included (1) history of skin disease; (2) fever (temperature ≥ 100.4 °F); (3) history of cardiovascular or pulmonary diseases; (4) adverse reactions to latex; (5) any known medical conditions that the principal investigator would not deem safe for participant; (6) inability or unwillingness to provide informed consent; (7) history of respiratory conditions; (8) medication usage that may cause methemoglobinemia; (9) history of smoking or narcotics usage; and (10) ongoing pregnancy. A total of 36 participants (mean age = 38.5 ± 16.5 years old, 23 females) signed written consent forms. After a participant signed the consent form, a physical screening, including a general health and medical history evaluation, an electrocardiogram, and pregnancy test for females of childbearing age, was performed by a nurse practitioner. Out of 36 participants, 26 (mean age = 34.7 ± 15.7 years old, 18 females) passed the physical screening and were enrolled in the study.

### 2.2. Experimental Procedures

Because the study was conducted during the COVID-19 pandemic, participants were required to wear a surgical mask throughout the measurement. All experimental procedures were performed in an outpatient clinic at the Clinical Center at the NIH. Before the experiment, measurement devices, including NIRS devices and other equipment, were shown and the experimental procedures were re-explained to participants. Participants were then asked if they wanted to continue participating. After a participant verbally confirmed their willingness to continue, an NIRS device was attached on participants’ forehead centered at Fpz location of the 10–20 international system using medical-grade, double-layer tape. During the experiment, participants sat comfortably on a hospital bed and were guided to follow instructions on a screen, which was approximately one meter away. The experiment began with a 5 min rest period, during which participants were asked to relax and breathe normally. A cross sign was displayed at the center of the screen during the first four minutes and a countdown timer was presented during the last minute. The rest period was followed by a breath-holding task. During the task, participants first exhaled and then held their breath for as long as they could within a maximum window of 2 min. The breath-holding task was repeated 3 times interspersed with 2 min rests. Once the breath-holding task was completed, the participants had a 5 min rest period.

### 2.3. Measurement Device

The hemodynamic responses were measured on participants’ forehead using a CW-NIRS device (prototype provided by Hamamatsu photonics K.K., Shizuoka, Japan). The NIRS device consists of a wearable, flexible probe and a portable controller. The probe has a multiwavelength Light-Emitting Diode (LED) operating at 730 nm, 800 nm, and 850 nm and two photodiode detectors that are placed 3 and 4 cm away from the LED. This arrangement forms two different source—detector separations (SDSs) (3 and 4 cm). The overall footprint of the probe is less than 40 cm^2^. The controller is connected to a cell phone for power supply and data recording. The cell phone is integrated with a mobile application to record and process data. Data were recorded at a sampling rate of 120 Hz. The mobile application outputs two worksheets, one for raw data and another for processed data. The processed data include change in oxyhemoglobin (HbO), deoxyhemoglobin (Hb), and total hemoglobin (THb) at each SDS (3 and 4 cm) and tissue oxygen saturation (StO_2_), which is calculated using raw data at both separations with the SRS method.

### 2.4. Data Analysis

Out of 26 recorded data sets, 7 data sets were discarded due to low signal to noise ratios. Among 19 participants with good data, 18 participants performed 3 breath-holding tasks and 1 participant performed 2 breath-holding tasks. Hence, there were total of 56 breath-holding tasks performed. The breath-holding time varied among participants (average time = 41.4 ± 22.4 s; range: 17.5–120 s).

In addition to StO_2_ obtained from the mobile application (StO_2_-SRS), separate StO_2_ values were calculated using the SSDS algorithm (StO_2_-SSDS) [37] (Equation (1)). Figure 1 illustrates the SRS method and SSDS algorithm. Backscattered light intensities and initial intensity ratios between wavelengths were extracted from the raw data worksheet. In this study, we used extinction coefficients of HbO and Hb at the three wavelengths reported in [38]. In addition, we assumed that the differential pathlength factor was the same for the three wavelengths (730 nm, 800 nm, and 850 nm).
(1)$${StO}_{2}=\frac{{\langle DPF\rangle }_{3}\cdot log(k_{1}.I_{\lambda_{2}}/I_{\lambda_{1}})\cdot \varepsilon_{\lambda_{3}}^{Hb}-{\langle DPF\rangle }_{1}\cdot log(k_{2}\cdot I_{\lambda_{2}}/I_{\lambda_{3}})\cdot \varepsilon_{\lambda_{1}}^{Hb}+{\langle DPF\rangle }_{2}\cdot log\left(k_{2}\cdot I_{\lambda_{1}}/k_{1}\cdot I_{\lambda_{3}}\right)\cdot \varepsilon_{\lambda_{2}}^{Hb}}{{\langle DPF\rangle }_{1}\cdot log\left(k_{2}\cdot I_{\lambda_{2}}/I_{\lambda_{3}}\right)\left(\varepsilon_{\lambda_{1}}^{HbO}-\varepsilon_{\lambda_{1}}^{Hb}\right)-{\langle DPF\rangle }_{3}\cdot log(k_{1}\cdot I_{\lambda_{2}}/I_{\lambda_{1}})\left(\varepsilon_{\lambda_{3}}^{HbO}-\varepsilon_{\lambda_{3}}^{Hb}\right)+{\langle DPF\rangle }_{2}\cdot log\left(k_{1}\cdot I_{\lambda_{3}}/k_{2}\cdot I_{\lambda_{1}}\right)\left(\varepsilon_{\lambda_{2}}^{HbO}-\varepsilon_{\lambda_{2}}^{Hb}\right)}$$
where ${\langle DPF\rangle }_{1}$ is the differential pathlength factor at wavelength $\lambda_{1}$, $k_{1}$ is the initial intensity ratio between wavelength $\lambda_{1}$ and $\lambda_{2}$, $I_{\lambda_{1}}$ is the backscattered light intensity at wavelength $\lambda_{1}$, and $\varepsilon_{\lambda_{1}}^{HbO}$ is the extinction coefficient of HbO at wavelength $\lambda_{1}$.

### 2.5. Statistical Analysis

Hemodynamic responses including HbO change, StO_2_-SRS, and StO_2_-SSDS were averaged for 5 s for statistical analysis. To eliminate baseline individual variation in hemodynamic responses, HbO change, StO_2_-SRS, and StO_2_-SSDS during and after the breath-holding task were subtracted from the baseline data (5 s before the task).

A single-factor one-way analysis of variance (ANOVA) test was performed on the baseline-subtracted, 5 s averaged hemodynamics responses before, during, and after the breath-holding task at different time points. By considering an averaged breath-holding time of 41.4 s, a time range from −5 s (5 s before the task) to 45 s was used. An ANOVA task yielding a *p* value equal to or less than 0.05 was considered statistically significant. Multiple one-sample, two-tailed post hoc *t*-tests (compared to zero) were performed on the baseline-subtracted, 5 s averaged hemodynamics responses during and after the breath-holding task following a significant ANOVA test result. In addition, Pearson correlation coefficients (ρ) among different hemodynamic responses and between the same hemodynamic response at different SDSs were calculated.

## 3. Results

### 3.1. HbO Change Due to the Breath-Holding Task

The averaged cerebral HbO changes at the forehead across all participants during and after the breath-holding task are presented in Figure 2. As previously described, the HbO change before, during, and after the task was subtracted from the baseline; hence, the mean and standard errors of HbO change at the baseline (from −5 to 0 s) are zero. During the first 10 s of the breath-holding task, the HbO change from both SDSs gradually decreases to −0.002 ± 0.004 µM (3 cm SDS) and −0.003 ± 0.005 µM (4 cm SDS). Between 10–15 s of the task, the HbO change starts increasing and reaches a value greater than the baseline HbO (mean HbO change = 0.005 ± 0.005 µM (3 cm SDS) and 0.007 ± 0.005 µM (4 cm SDS)) at 15–20 s. After that, the HbO change continues to increase until it reaches the maximal peak (mean HbO change = 0.021 ± 0.005 µM (3 cm SDS) and 0.023 ± 0.005 µM (4 cm SDS)) at 25–30 s and then eventually decreases toward the baseline HbO. At 40–45 s, the HbO change is 0.010 ± 0.005 µM (3 cm SDS) and 0.008 ± 0.005 µM (4 cm SDS) above the baseline HbO. A correlation analysis demonstrates that HbO changes from the two SDSs are highly correlated (ρ = 0.99). Single-factor one-way ANOVA tests on the baseline-subtracted, 5 s averaged HbO change across 56 breath-holding tasks indicate a significant difference among different times for both 3 cm (*F*(9) = 3.7, *p* < 0.001) and 4 cm (*F*(9) = 4.8, *p* < 0.001) SDSs. Post hoc one-sample *t*-tests (compared to zero) show that the HbO change is significantly greater than zero at 20–25 s (*p* = 0.02 (3 cm SDS) and *p* = 0.005 (4 cm SDS)), 25–30 s (*p* < 0.001 (3 and 4 cm SDSs)), 30–35 s (*p* = 0.002 (3 cm SDS) and *p* < 0.001 (4 cm SDS)), and 35–40 s (*p* = 0.007 (3 cm SDS) and *p* = 0.005 (4 cm SDS)).

### 3.2. Cerebral StO_2_-SRS Change Due to the Breath-Holding Task

Similar to HbO, the cerebral StO_2_-SRS initially decreases during the first 10 s (mean StO_2_-SRS change = −0.10 ± 0.17%) and gradually increases after that (Figure 3). Between 15–20 s of the task, the cerebral StO_2_-SRS is greater than the baseline StO_2_ (mean StO_2_-SRS change = 0.24 ± 0.19%). The maximum StO_2_-SRS change is 0.48 ± 0.21% at 25–30 s. Eventually, the StO_2_-SRS decreases toward the baseline StO_2_-SRS. At 40–45 s, the StO_2_-SRS is 0.24 ± 0.23% below the baseline StO_2_-SRS. A single factor one-way ANOVA test on the baseline-subtracted, 5 s averaged StO_2_-SRS across 56 breath-holding tasks indicates a significant difference among different times (*F*(9) = 3.3, *p* < 0.001). Post hoc one-sample *t*-tests (compared to zero) show that StO_2_-SRS is significantly greater than zero at 25–30 s (*p* = 0.03). A correlation analysis between the HbO change and StO_2_-SRS change shows that the StO_2_-SRS change is greater correlated to the HbO change from 4 cm SDS (ρ = 0.81) than the HbO change from 3 cm SDS (ρ = 0.73).

### 3.3. Cerebral StO_2_-SSDS Change Due to the Breath-Holding Task

Due to a low signal to noise ratio, data at 4 cm SDS from one participant were excluded. As a result, StO_2_-SSDS at 3 cm SDS consists of data from 56 breath-holding tasks, but StO_2_-SSDS at 4 cm SDS consists of data from 53 breath-holding tasks. Figure 4 represents cerebral StO_2_-SSDS change in response to the breath-holding task, which has a similar trend to HbO and StO_2_-SRS. Cerebral StO_2_-SSDS from both SDSs gradually decreases during the first 10 s to −0.04 ± 0.4% (3 cm SDS) and −0.06 ± 0.3% (4 cm SDS). It then steadily increases to the maximal peak at 25–30 s (mean StO_2_-SSDS change = 1.22 ± 0.56% (3 cm SSDS) and 0.82 ± 0.33% (4 cm SSDS)). After that, it eventually decreases toward the baseline StO_2_-SSDS. At 40–45 s, StO_2_-SSDS is 0.54 ± 0.58% above the baseline StO_2_-SSDS (3 cm SDS) and 0.17 ± 0.39% below the baseline StO_2_-SSDS (4 cm SDS). An analysis between HbO and StO_2_-SSDS indicates a higher correlation between the two parameters at 3 cm SDS (ρ = 0.99) than at 4 cm SDS (ρ = 0.87). The correlation coefficients between HbO and StO_2_-SSDS are higher than the ones between HbO and StO_2_-SRS. An analysis between StO_2_-SRS and StO_2_-SSDS shows that StO_2_-SRS is more correlated with StO_2_-SSDS at 4 cm SDS (ρ = 0.98) than with StO_2_-SSDS at 3 cm SDS (ρ = 0.78).

### 3.4. Cerebral StO_2_ during a Long versus Short Breath-Holding Task

Figure 5 shows cerebral StO_2_ at the forehead during a breath-holding task in two representative participants, who were able to hold their breath for 25 s (duration indicated by vertical lines, Figure 5a) and 120 s (duration indicated by vertical lines, Figure 5b). In both participants, cerebral StO_2_ initially decreased to a minimal level and then gradually increased to an StO_2_ level greater than the resting StO_2_ at the end of the task. However, in the participant with a short hold (25 s), StO_2_ was decreasing continuously for approximately 5 s, but in the participant with a long hold (120 s), StO_2_ was decreasing continuously for approximately 15 s. In addition, after reaching its minimal peak, StO_2_ steadily increased until the end of the task in a short hold, but in a long hold, StO_2_ fluctuated greatly. Comparing between StO_2_-SRS (yellow trace) and StO_2_-SSDS (blue and red traces), StO_2_ alteration in response to a long hold is similar between SRS and SSDS (Figure 5). However, StO_2_ alteration in response to a short hold is more observable in StO_2_-SSDS than in StO_2_-SRS. Specifically, at 5 s, StO_2_-SSDS decreases by 0.84% (3 cm SDS) and 0.94% (4 cm SDS), while StO_2_-SRS decreases only 0.51% from the resting StO_2_. Moreover, at the end of the task, StO_2_-SSDS increases 2.71% (3 cm SDS) and 1.65% (4 cm SDS), while StO_2_-SRS increases only 0.03% from the resting StO_2_. The small change in StO_2_-SRS in response to a short breath-holding task makes it indistinguishable from background variation during rest. Furthermore, StO_2_-SRS has lower signal to noise ratio compared to StO_2_-SSDS at both SDSs and StO_2_-SRS shares the same artifacts (spikes) with StO_2_-SSDS at 4 cm SDS, in both the short and long breath-holding tasks (Figure 5).

## 4. Discussion

In this study, we measured cerebral hemodynamic responses, including HbO change and StO_2_, during a breath-holding task. Cerebral StO_2_ was derived from two different methods, namely SRS and SSDS. We have shown that both HbO change and StO_2_ initially decrease to a minimal peak at around 10 s of the breath-holding task and then steadily increase to a maximum peak, which is significantly greater than the baseline level, at 25–30 s. Comparing between SRS and SSDS, we have demonstrated that SSDS is more robust than SRS in quantifying changes in cerebral StO_2_ in response to a breath-holding task. In addition, we found that StO_2_-SSDS has a higher correlation with HbO change than StO_2_-SRS.

Our result of hemodynamic responses to a breath-holding task is consistent with previous studies, in which controlled breath holding was used to model the systemic physiological effects of hypoxia in patients [39,40]. The act of breath holding leads to a decrease in the partial pressure of oxygen and a rise in carbon dioxide (CO_2_) above the normative value of around 40 mmHg [40]. This change can damage vital organs such as the brain. As a result, to protect the brain, a compensatory neurovascular response occurs to provide a constant supply of oxygen to the brain through increased cerebral blood flow [1]. This increased neurovascular blood flow is achieved through dilation of intracranial and extracranial blood vessels [1,2,3], particularly in the middle cerebral artery which supports this compensatory response [21]. However, the response can only be maintained for a limited time as neural tissue death can occur in under 3 min when the supply of oxygen is interrupted [4]. The hypoxic responses to breath holding can happen in as little as 5 s where peripheral oxygen saturation drops continuously. In contrast, the cerebral tissue oxygen saturation will immediately drop at the start of the event but subsequently recovers to a level slightly above the baseline [39].

Even though there is an observable trend in hemodynamic responses to the breath-holding task and the trend is similar to previous studies, there is a large standard error throughout the task. In particular, the standard error tends to increase toward the end of the task. The large standard error could be because of the difference in individual hemodynamic responses to the task and breath-holding duration between participants. For example, the cerebral StO_2_ reaches its minimal peak at an earlier time and with a smaller amplitude in a short hold compared to a long hold. In addition, from 5 s to 15 s, StO_2_ is continuously decreasing during a long hold, but it is steadily increasing in a short hold. In contrast to the change at the beginning of the breath-holding task, the cerebral StO_2_ increases and reaches its maximal peak at the end of the task. Thus, after 25 s, cerebral StO_2_ starts decreasing towards the baseline in a short hold, but it continues to increase in a long hold.

The results of this study indicate that SSDS is better than SRS in monitoring changes in cerebral StO_2_ in response to a breath-holding task. Firstly, SSDS is able to more accurately detect change in cerebral StO_2_ in response to a short breath-holding task, while SRS fails to do so. Secondly, StO_2_-SSDS at 4 cm SDS is highly correlated with StO_2_-SRS and StO_2_-SRS shares the same artifacts with StO_2_-SSDS at 4 cm SDS. Due to the far distance between a light source and a photodetector at 4 cm SDS, both SRS and SSDS at 4 cm SDS suffers from a low signal to noise ratio, especially in a tissue with high absorption. Thirdly, due to an assumption that the distance between photodetectors is substantially smaller than the distance between the light source and photodetectors in SRS, photodetectors often should be placed far away from the light source. As a result, SRS cannot be used to quantify oxygen saturation at superficial tissues, making contamination from the scalp and skull unavoidable. On the other hand, SSDS can be applied on a close SDS (for example SDS < 1 cm) to calculate the StO_2_ of superficial tissues, which can be used to eliminate the effect of scalp and skull StO_2_ on measured cerebral StO_2_. Finally, since SSDS requires information from only one source—detector pair, devices using SSDS can be made smaller and more cost-effective.

Tissue oxygen saturation is a mixture of blood oxygen saturation in arteries, veins, and capillaries. Therefore, depending on the distribution of blood vessels, StO_2_ can vary greatly between people. As a result, StO_2_ cannot be used to diagnose hypoxia. However, it can be used in a monitoring manner for daily use or neurosurgery.

## 5. Conclusions

This study applies the single source—detector separation algorithm to calculate cerebral tissue oxygen saturation during a breath-holding task. We have proved that this algorithm is better than the conventional spatially resolved spectroscopy method in detecting and quantifying alterations in cerebral tissue oxygen saturation in response to a breath-holding task. In addition, because the single source—detector separation algorithm requires information from only one source—detector pair, it can be a great candidate for a miniaturized biosensor to monitor tissue oxygen saturation as well as to detect hypoxia. The miniaturized biosensor can not only be used as a standalone wearable device for monitoring tissue oxygen saturation noninvasively but it can also be attached on an endoscope for measuring the oxygen saturation of internal organs. The wearable device can be used to detect hypoxia in divers, people living at high altitudes, and patients undergoing neurosurgery. The endoscopic biosensor can be used to study the oxygen saturation of inflamed areas or cancer tissues.

## Figures and Tables

**Figure 1 bioengineering-11-00385-f001:**
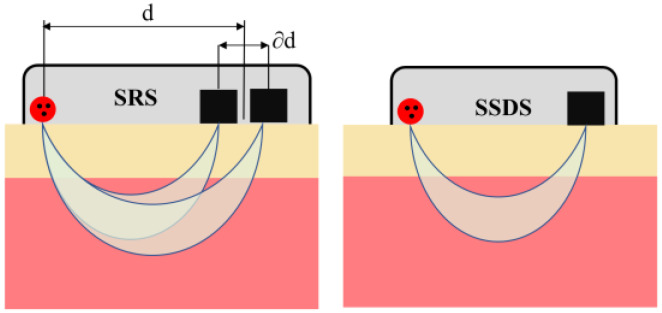
Illustration of devices that use the SRS method and SSDS algorithm. SRS-based devices need to have at least 2 source—detector pairs, while SSDS-based devices need only one source—detector pair; hence, they can be made smaller and simpler. In addition, there is an assumption that d has to be significantly larger than ∂d in the SRS method. As a result, SRS-based devices need to have detectors far away from the light source, which reduces the signal to noise ratio and increases the device size.

**Figure 2 bioengineering-11-00385-f002:**
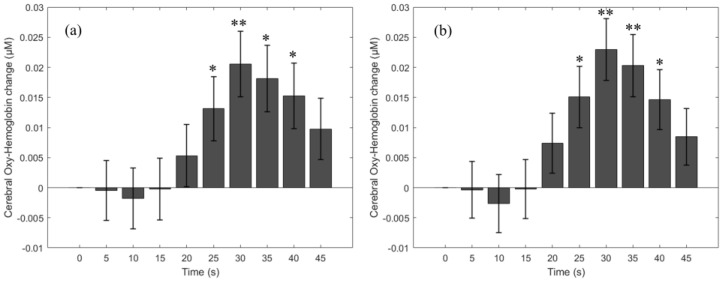
Cerebral HbO changes due to the breath-holding task; (**a**) HbO change from 3 cm SDS; (**b**) HbO change from 4 cm SDS. Error bars represent standard error. * indicates a *p* value < 0.05 and ** indicates a *p* value < 0.001.

**Figure 3 bioengineering-11-00385-f003:**
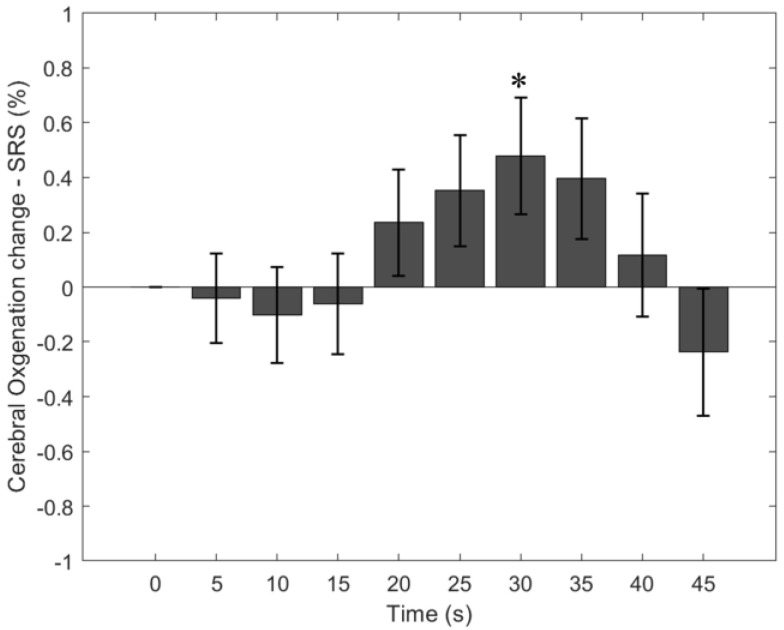
Change in StO_2_-SRS due to the breath-holding task. Error bars represent standard error. * indicates a *p* value < 0.05. StO_2_-SRS change was obtained from the processed data worksheet.

**Figure 4 bioengineering-11-00385-f004:**
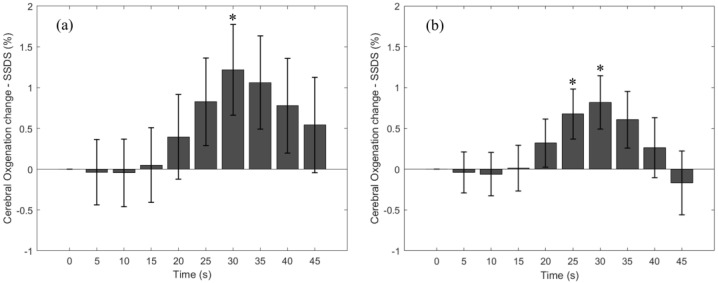
Change in StO_2_-SSDS due to the breath-holding task. (**a**) StO_2_-SSDS calculated from 3 cm SDS; (**b**) StO_2_-SSDS calculated from 4 cm SDS. Error bars represent standard error. * indicates a *p* value < 0.05.

**Figure 5 bioengineering-11-00385-f005:**
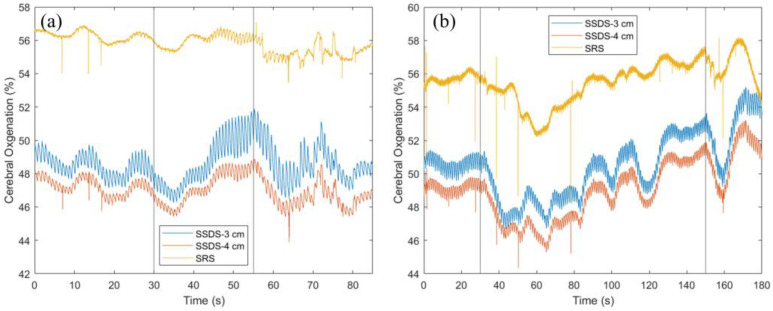
StO_2_-SRS and StO_2_-SSDS before, during, and after a breath-holding task; (**a**) short task and (**b**) long task. Vertical lines mark the start and end of the breath-holding task.

## Data Availability

Data supporting the reported results will be uploaded to Github upon publication.

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
