# Peer review of "Application of the Single Source—Detector Separation Algorithm in Wearable Neuroimaging Devices: A Step toward Miniaturized Biosensor for Hypoxia Detection"

_bioengineering, 2024, doi:10.3390/bioengineering11040385_

Round 1

Reviewer 1 Report

Comments and Suggestions for Authors

Thien Nguyen et al. reported an interesting work about an algorithm in wearable neuroimaging devices. The strategy is useful for future clinical diagnostics. The topic was of a certain significance, and the paper could be considered for publication in Bioengineering. Thus, the reviewer suggested a Minor Revision for this paper. Please refer to the following comments:

1.       At the end of the Introduction, please consider to add a scheme to illustrate the advancement of the current work over the previous one of SSDS.

2.       The mechanisms of the algorithm should be discussed in detail in the Discussion.

3.       As indicated in the Title, the findings should be useful for wearables. Could the authors show an example of a wearable device using the algorithm?

4.       The clinical translation aspects could be envisioned in the Conclusions.

5.       Please comment on whether the algorithm could be used for clinical imaging?

Author Response

We would like to extend our sincere gratitude to the reviewer. Thank you so much for your work and supportive comments. Please find our responses to your comments in the attached file.

Reviewer 2 Report

Comments and Suggestions for Authors

The authors may consider including an illustration of two-pair versus single-pair detector systems to emphasize how the latter is suitable for a potential wearable sensor. 

Author Response

Thank you so much for your suggestion. We added Figure 1 to the manuscript to illustrate the advantages of SSDS over SRS. Please find the added figure in the attached file or in the revised manuscript.
